Evaluation of AAV-DJ vector for retinal gene therapy

Katada Yusaku 1 2
Kobayashi Kenta 3
Tsubota Kazuo tsubota@z3.keio.jp 1
Kurihara Toshihide kurihara@z8.keio.jp 1 2
1 Department of Ophthalmology, School of Medicine, Keio University , Shinjuku-ku , Tokyo , Japan
2 Laboratory of Photobiology, School of Medicine, Keio University , Shinjuku-ku , Tokyo , Japan
3 Section of Viral Vector Development, Center for Genetic Analysis of Behavior, National Institute for Physiological Sciences, National Institutes of Natural Sciences , Okazaki , Aichi , Japan
Yusa Keisuke
Electronic publication date: 2019 Jan 17
Publication date: 2019
Volume: 7
Electronic Location ID: e6317
Received 2018 Oct 15; Accepted 2018 Dec 19
Copyright: ©2019 Katada et al.
Copyright year: 2019
Copyright holder: Katada et al.
License: This is an open access article distributed under the terms of the Creative Commons Attribution License, which permits unrestricted use, distribution, reproduction and adaptation in any medium and for any purpose provided that it is properly attributed. For attribution, the original author(s), title, publication source (PeerJ) and either DOI or URL of the article must be cited.
License URL: https://creativecommons.org/licenses/by/4.0/

Keywords: Adeno-associated viral vector, Gene therapy, Retina

Funding: Keio University Doctorate Student Grant-in-Aid Program Takeda Science Foundation Keio University Medical Science Fund This work was supported by grants from the Keio University Doctorate Student Grant-in-Aid Program, the Takeda Science Foundation and the Keio University Medical Science Fund. The funders had no role in study design, data collection and analysis, decision to publish, or preparation of the manuscript.

==============================
Purpose

The most common virus vector used in gene therapy research for ophthalmologic diseases is the adeno-associated virus (AAV) vector, which has been used successfully in a number of preclinical and clinical studies. It is important to evaluate novel AAV vectors in animal models for application of clinical gene therapy. The AAV-DJ (type 2/type 8/type 9 chimera) was engineered from shuffling eight different wild-type native viruses. In this study, we investigated the efficiency of gene transfer by AAV-DJ injections into the retina.

Methods

One microliter of AAV-2-CAGGS-EGFP or AAV-DJ-CAGGS-EGFP vector at a titer of 1.4 × 10e12 vg/ml was injected intravitreally or subretinally in each eye of C57BL/6 mice. We evaluated the transduction characteristics of AAV-2 and -DJ vectors using fluorescence microscopy and electroretinography.

Results

The results confirmed that AAV-DJ could deeply transfer gene to photoreceptor layer with intravitreal injection and has an efficient gene transfer to various cell types especially the Mueller cells in the retina. Retinal function was not affected by AAV-DJ infection or ectopic EGFP expression.

Conclusions

The AAV-DJ vector efficiently induces the reporter gene in both the inner and outer murine retina without functional toxicity. These data indicated that the AAV-DJ vector is a useful tool for the gene therapy research targeting retinal disorders.

Introduction

No fundamental therapies exist for retinal degenerative diseases such as retinitis pigmentosa and dry age-related macular degeneration, although numerous surgical and medical innovations have been introduced to treat cataract and glaucoma, which were the leading causes of blindness worldwide. Establishment of efficient therapeutic approaches for degenerative retinal diseases is strongly desired. To date, gene therapy is a promising candidate to prevent blindness in these patients.

Currently, the most common virus vector used in gene therapy research for ophthalmologic diseases is the adeno-associated virus (AAV) vector, which has been used in a number of successful preclinical and clinical studies (Bainbridge et al., 2008; Hauswirth et al., 2008; Maguire et al., 2008; Jacobson et al., 2012; MacLaren et al., 2014; Hoxha et al., 2016). Since AAV vectors can transduce genes into non-dividing cells such as neurons, it is advantageous for tissues such as the retina. As a virus, AAV is an excellent vector since it has low cytotoxicity and pathogenicity. In December 2017, the FDA approved the first gene therapy product using AAV for an inherited retinal disease (Russell et al., 2017; Miraldi Utz et al., 2018).

To date, about 12 types of serotypes have been reported for AAV vectors (Schmidt et al., 2008b; Schmidt et al., 2008a), and each serotype has its own external capsid protein. Tissue tropism is determined by the serotype of the AAV vector and is therapeutically important. Type 2 AAV (AAV-2), the first vector used in gene therapy, is well characterized and widely used in Ophthalmology (Bainbridge et al., 2008; Hauswirth et al., 2008; Maguire et al., 2008).

Vector systems based on other AAV serotypes with more efficient gene transduction and different cell and tissue specificities are being investigated. In addition to the naturally occurring series of AAVs, novel recombinant AAV capsid variants with efficient gene transfer and tropism have been generated recently either by rational design or directed evolution (Carvalho & Vandenberghe, 2015). It is important to continuously evaluate novel AAV vectors for human gene therapy and its basic research, because cell-type-specific gene introduction tailored to each disease will make treatment more efficient and minimize the immunologic sequelae, and increased packaging capacity expands therapeutic gene options (Taymans et al., 2007; Zaiss & Muruve, 2008). Although AAV-2 is currently most frequently used, the serotype cannot introduce genes into photoreceptors with intravitreal injection (Dalkara et al., 2009; Yin et al., 2011). Intravitreal injection is less invasive and a route of administration performed routinely for clinical human subjects. New AAV capsids that penetrate photoreceptors with intravitreal injection can be a revolution in the retinal gene therapy.

The AAV-DJ (type 2/type 8/type 9 chimera), which was engineered from shuffling eight different wild -native viruses (Grimm et al., 2008), has been used widely for gene transduction due to the highly efficient gene transfer into a broad range of cell types. For example, the AAV-DJ vector was used to knock out a gene in porcine fibroblasts with a higher targeting frequency than the other naturally occurring serotypes (Hickey et al., 2011). AAV-DJ greatly outperformed all other serotypes at transduction efficiency in human keratinocytes (Melo et al., 2014). In the current study, we investigated the efficiency of gene transfer to the retina by AAV-DJ to determine the usability for clinical applications.

Materials & Methods

Animals

All animal experiments were conducted in accordance with protocols approved by Institutional Animal Care and Use Committee of Keio University School of Medicine (#2808). Only adult C57BL/6 mice were used in this study. All animals were maintained under a 12∕12-h light/dark environment.

Vector production and purification

pAAV-CAGGS-EGFP-WPRE was used. The CAGGS promoter contains a cytomegalovirus enhancer and a chicken β-actin promoter (Niwa, Yamamura & Miyazaki, 1991). All serotypes of AAV vectors were prepared using AAV Helper Free Packaging System (Cell Biolabs, San Diego, CA, USA). The serotypes were produced in HEK 293 cells using a helper virus-free system and purified using two CsCl2 density gradients and titrated by quantitative polymerase chain reaction. Final preparations were dialyzed against phosphate-buffered saline (PBS) and stored at −80 °C.

Virus injection

The mice were anesthetized with pentobarbital sodium at the dose of 50 mg/kg body weight and placed on a heating pad that maintained their body temperature at 35–36 °C throughout the experiments. An aperture was made next to the limbus through the sclera with a 30-gauge disposable needle, and a 33-gauge unbeveled blunt-tip needle on a Hamilton syringe was introduced through the scleral opening into the vitreous space for intravitreal injections and introduced through the scleral opening along the scleral interior wall into the subretinal space for subretinal injections. Each eye received 1 µl of vehicle (PBS) or vector at a titer of 1.4 × 10e12 vg/ml.

Fundus photography

The mice were anesthetized with pentobarbital sodium at the dose of 50 mg/kg body weight and the pupils were dilated with a mixed solution of 0.5% tropicamide and 0.5% phenylephrine (Mydrin-P; Santen, Osaka, Japan). Live fundus images after the AAV injection were captured using a stereomicroscopy (M165-FC; Leica Microsystems, Wetzlar, Germany) with a glass slide and water to cancel the refraction. Fluorescence intensity was measured from photographs taken under the same conditions using ImageJ software (National Institutes of Health, Bethesda, MD, USA).

Preparation of retinal flat mounts and cryosections

Two months after vector injection, the mice were humanely euthanized. The eyes were removed and fixed with 4% paraformaldehyde in PBS for 1 h, and the cornea and lens were removed. To make flat mounts, the entire retina was dissected carefully from the eyecup and radial cuts were made from the edges to the retinal equator. For cryosections, the eyecups were washed in PBS followed by immersion in 30% sucrose in the same buffer overnight. The eyecups then were cryoprotected in O.C.T. Compound (Sakura Finetek Japan, Tokyo, Japan) and cryosectioned into 12-µm-thick transverse retinal sections.

Immunohistochemistry

Transduced mouse retinas were dissected and fixed in 4% paraformaldehyde for 30 min at room temperature. The retinas were incubated in PBS with 1% Triton X-100, 0.5% Tween 20 for 1 h at room temperature and in 10% normal goat serum for 1 h at room temperature and then incubated overnight at 4 °C with primary antibody: Brn3a (1:100; Santa Cruz Biotechnology, Santa Cruz, CA, USA), rhodopsin (1:500; Abcam, Cambridge, UK), PKCα (1:500; Abcam), calbindin (1:500; Abcam) and glutamine synthetase (GS) (1:500; Abcam) in blocking buffer. Secondary anti-rabbit IgG and anti-mouse IgG conjugated with Alexa TM594 and Alexa TM555, respectively (1:1,000; Molecular Probes), were applied for 1 h at room temperature. The retinas then were flat-mounted and the sections mounted on slide glass.

Fluorescent imaging

The retinal flat-mount images were obtained using a fluorescence microscope (BZ-9000, Keyence, Osaka, Japan). Fluorescent images of the cryosections were obtained using a confocal microscope (LSM710, Carl Zeiss, Jena, Germany).

ERG analyses

The scotopic ERGs were recorded according to a previous report (Zheng et al., 2012). The animals were dark-adapted for 12 h and prepared under dim red illumination. Damaged control mice with light-induced retinopathy were prepared according to a previous report (Kubota et al., 2010). The mice were anesthetized with a combination of midazolam, medetomidine, and butorphanol tartrate at the doses of 4, 0.75, and 5 mg/kg body weight and placed on a heating pad that maintained their body temperature at 35–36 °C throughout the experiments. The pupils were dilated with a mixed solution of 0.5% tropicamide and 0.5% phenylephrine (Mydrin-P, Santen, Osaka, Japan). The ground and references electrodes were placed subcutaneously into the tail root and between the eyes, respectively. The active contact lens electrodes (Mayo, Inazawa, Japan) were placed on the cornea. Recordings were performed using the PuRec system (Mayo) and filtered through a bandpass filter ranging from 1 to 100 Hz to yield the a- and b-waves. Light pulses of 2.0 log cd-s/m2 were delivered via a full-field light stimulator LS-100 (Mayo).

Statistical analysis

Statistical analysis was performed using SPSS software version 25 for Windows (IBM, Armonk, NY, USA). The criterion for statistical significance was p < 0.05, and the unpaired t-test or analysis of variance was performed.

Results

Time course of enhanced green fluorescent protein (EGFP) expression in murine retina after AAV intravitreal injection

The purified AAV-2-CAGGS-EGFP or AAV-DJ-CAGGS-EGFP virus vector (hereafter referred to as AAV-2 or AAV-DJ) at a titer of 1.4 × 10e12 vg/ml was injected intravitreally and the changes over time in the EGFP signals in the retinas of live mice were observed using a fluorescence stereoscopic microscope. A fluorescent signal was confirmed 1 week after the injection and peaked at 8 weeks after the injection in both serotypes (Fig. 1A). AAV-DJ showed a rapid rise in fluorescence intensity, and its fluorescence intensity (n = 6 retinas) was significantly higher at 1 week after injection than that of AAV-2 (n = 6 retinas; p < 0.001, unpaired t-test) (Fig. 1B). Thereafter, the fluorescence intensity of AAV-2 (n = 6 retinas) also increased and tended to be higher than AAV-DJ at 4 weeks (n = 6 retinas; p = 0.11, unpaired t test) (Fig. 1C). We then evaluated the strength and efficacy of gene expression in retinal ganglion cells (RGCs) by investigating localization of expression by co-labeling with Brn3a, a reliable marker to identify and quantify RGCs (Quina et al., 2005). Confocal microscopy images of retinal flat mounts showed strong EGFP expression in RGCs and this expression was co-localized with the Brn3a labeling (Figs. 1D, 1E). Cell quantification in such images indicated that 81 ± 12.2% of these Brn3a-positive RGCs also expressed EGFP in the AAV-2 injected retinas (n = 24 images, n = 6 retinas) compared with 67 ± 72.6% in the AAV-DJ injected retinas (n = 24 images, n = 6 retinas; p = 0.095, unpaired t-test) (Fig. 1F).

Figure 1 Time course of EGFP expression in the retina following AAV intravitreal injection.

(A, B) Representative fundus photographs show EGFP expression in live mice from 1 through 12 weeks after the AAV injection. A: AAV-2-CAGGS-EGFP, B: AAV-DJ-CAGGS-EGFP. (C, D) Quantification of EGFP fluorescence intensity from fundus fluorescent photographs. The mean fluorescence values were measured over the circular imaging range. (E, F) Representative confocal image across the RGC layer of flat mounted mouse retina transduced with AAV-2-CAGGS-EGFP or AAV-DJ-CAGGS-EGFP at 4 weeks after injection, co-labeled with Brn3a (red). (G) Quantification of the EGFP-positive rate in Brn3a-positive cells. The cells were counted in a region 212-µm-square on each top, bottom, left, and right side of the flat-mounted retina and averaged. Error bars represent the standard error of the mean. Scale bars, 50 µm in E and F. *** p < 0.001 unpaired t-test. ns, not significant.

Histologic comparison of EGFP expression mediated by AAV-2 vs. AAV-DJ

Two months after the intravitreal injection, EGFP expression was observed across the ganglion cell layer (GCL) side of the retinal flat mounts with both serotypes (Figs. 2A, 2C). Uniform EGFP expression was observed in the retinas injected with AAV-2 (Fig. 2A). In contrast, accumulation of EGFP signaling was observed with AAV-DJ injection in particular areas such as around the blood vessels covered with Mueller glia’s protrusions (Figs. 2E, 2F, Fig. S1). Cross-sections and immunohistochemical analysis showed EGFP fluorescence in RGC cell soma and axons, amacrine cells and horizontal cells in both serotypes (Figs. 2B, 2G, Fig. S2), whereas expression in Mueller cells (Figs. 2D, 2J–2L), bipolar cells, and rod photoreceptors (Figs. 3A–3C) was only found with AAV-DJ.

Figure 2 EGFP expression into the murine retina with intravitreal injection of AAV-DJ vector.

Representative confocal images of (A, C, E, F) whole-mount and (B, D,G) cross-sections of retinas 2 months after (A, B) AAV-2-CAGGS-EGFP intravitreal injection and (C–G) AAV-DJ-CAGGS-EGFP intravitreal injection. DAPI nuclear counterstain is shown in blue. (E, F) A high-magnification view of the optic disc and retinal vessels; asterisks indicate retinal vessels. (G) A high-magnification view of the optic nerve head; arrowheads indicate EGFP-positive axons converging from the inner retinal surface to the optic nerve. INL, inner nuclear layer; ONL, outer nuclear layer; ON, optic nerve. Scale bars, 500 µm in A, C, E and 50 µm in B, D, E, G.

Figure 3 Immunostaining of each retinal cell type of AAV-DJ intravitreally injected mouse retina.

Immunohistochemistry on transverse retinal cryosections from murine retinas with intravitreal injection of AAV-DJ-CAGGS-EGFP for (A–C) rod photoreceptors labelled with rhodopsin, (D–F) bipolar cells labelled with PKCα, (G–I) horizontal cells labelled with calbindin, and (J–L) Mueller cells labelled with GS. INL, inner nuclear layer; ONL, outer nuclear layer. Scale bars, 50 µm in A–L.

However, EGFP expression was detectable only around the injection site after subretinal injection of both AAV serotypes (Fig. 4). In the cross-sections, EGFP expression was limited mainly to the photoreceptor outer segment (POS) with AAV-2 injections. In contrast, the expression spread into the entire photoreceptor layer, and a few signals in the inner retinal layers also were seen with AAV-DJ injections (Figs. 4D–4G).

Figure 4 EGFP transfection into the murine retina with subretinal injection of AAV-DJ vector.

Representative confocal images of (A, C) whole-mounts and (B, D) cross-sections of retinas 2 months after (A, B) AAV-2-CAGGS-EGFP subretinal injection and (C, D) AAV-DJ-CAGGS-EGFP subretinal injection. (E–G) Immunohistochemistry on transverse retinal cryosections from murine retinas with subretinal injection of AAV-DJ-CAGGS-EGFP for rod photoreceptors labelled with rhodopsin. DAPI nuclear counter stain is shown in blue. INL, inner nuclear layer; ONL, outer nuclear layer. Scale bars, 500 µm in A, C, and 50 µm in B, D, E–G.

Functional changes after AAV injection

To examine changes in retinal function associated with AAV infection, electroretinography (ERG) was performed. The scotopic ERG showed a slight but non-significant reduction in the b-wave amplitude with injection of vehicle (n = 6 eyes), AAV-2 (n = 6 eyes), or AAV-DJ (n = 6 eyes) compared to age-matched controls without injections (n = 6 eyes). However, no significant difference was observed between the injection groups (Fig. 5). These data indicated that the reduction in the ERG amplitude was not due to infectious toxicity of AAV but invasion of the intravitreal injection itself.

Figure 5 ERG results obtained from AAV intravitreally injected mice.

(A–E) Representative ERG waveforms in 10-week-old mice. (A) age-matched control mice, (B) 1 month after PBS intravitreal injection, (C) light-induced damaged control mice, (D) 1 month after AAV-2-CAGGS-EGFP intravitreal injection, (E) AAV-DJ-CAGGS-EGFP intravitreal injection. (F) The average a-wave amplitude in the scotopic ERG. (G) The average b-wave amplitude in the scotopic ERG. Error bars represent the standard error of the mean. ns, not significant. ***p < 0.001 analysis of variance.

Discussion

It has been reported that AAV-DJ showed significantly higher infection efficiency than a native AAV serotype for a wide range of tissues and cell types (Grimm et al., 2008). In the current study, we found that AAV-DJ had high gene transfer efficiency with all types of retinal cells in vivo. The two common approaches of intraretinal gene delivery are intravitreal and subretinal injections (Liang et al., 2001). The EGFP signal transfected by the AAV-DJ intravitreal injection was observed from the GCL into the POS. In particular, EGFP expression was observed in Mueller cells, and the expression along the blood vessels seen in flat-mount retinas was Mueller glia’s protrusions. Astrocytes may also be involved due to the cellular similarity as previously reported (Klimczak et al., 2009). Few serotypes exhibit efficient transduction by intravitreal injection, because the inner limiting membrane (ILM) has been implicated as the barrier responsible for the inefficiency of most rAAV vectors in transducing the retina (Dalkara et al., 2009). It is known that heparin affinity of AAV capsid plays an important role in infection efficiency into the retina in intravitreal injection (Wu et al., 2006; Hellström et al., 2009; Boye et al., 2016; Woodard et al., 2016). AAV-DJ contains the heparin binding domain from AAV-2, and the electrostatic potential overall is quite similar to that of AAV-2 (Lerch et al., 2012). Therefore, it can be considered that AAV-DJ showed efficient introduction by intravitreal injection. In addition, the high expression efficiency in Mueller cells is considered to be due to the first contact of intravitreal AAV occurring at ILM, the end feet of Mueller glia. Therefore, the early rising of the fluorescence intensity (Figs. 1A–1C) might reflected the expression of EGFP in Mueller cells.

There are some AAV variants also reported to transfer gene into the retinal pigmented epithelium by intravitreal injection. These reports described that the increased retinal transduction efficiency might arise from the variant’s reduced heparin affinity that cause both decreased capsid sequestration in the ILM and enhanced penetration through retinal layers (Dalkara et al., 2013; Kay et al., 2013). Like their variants, it is noteworthy that AAV-DJ is also an AAV2-based variant and binds heparin but with lower affinity than that of wild-type AAV2 (Grimm et al., 2008). AAV-DJ can be used for gene therapy of various cell types. In particular, it might be suitable for gene therapy targeting Mueller cells (Pellissier et al., 2014; Pellissier et al., 2015; Byrne et al., 2014) or retinal vascular lesions such as anti-angiogenic therapy (Pechan et al., 2009; Paulus & Sodhi, 2016) because of the expression along the retinal vessels (Figs. 2E, 2F, Fig. S1). Since AAV-DJ has a different VR-1 conformation from AAV-2, the resistance to human polyclonal neutralization is also advantageous (Grimm et al., 2008; Lerch et al., 2012).

Regarding the safety of AAV, viral infection and the intracellular presence of the viral vector potentially induces cellular stress and toxicity, and the transgene might affect retinal function. AAV vector particles can persist in the photoreceptors for an extended time after subretinal injection (Rex et al., 2005). However, previous studies have shown that high intracellular concentrations of transfected GFP did not have toxic effects (Daniels et al., 2003; Rex et al., 2005). Indeed, there were no changes in the ERGs between intravitreal vehicle and AAV injection, and no toxicity in the EGFP expression or AAV-DJ infection that affected the retinal function was observed (Fig. 5).

Conclusions

The current study showed a high infection efficiency of AAV-DJ for a wide range of retinal cell types. Using AAV-DJ, intravitreal injections facilitated gene transfer to the depth of the photoreceptors throughout the entire retina, especially the Mueller cells, and subretinal injection allowed gene transfer into the outer retina locally but efficiently. Additional information on transduction and the cellular tropism characteristics of different AAV vectors might help in the development of improved protocols for treating various types of retinal disease. The AAV-DJ vector has practical possibilities for retinal research and gene therapy for the outer and inner retina and especially retinal vascular disorders.

Supplemental Information

Figure S1 Immunostaining around the blood vessel of AAV-DJ intravitreally injected mouse retina

Immunohistochemistry for GS (red) labelling Mueller cells on the transverse retinal cryosection with intravitreal injection of AAV-DJ-CAGGS-EGFP (green). Note that processes of Mueler glia around the blood vessel co-labelled with GS and EGFP. The dotted line indicates a retinal blood vessel. INL, inner nuclear layer; ONL, outer nuclear layer. Scale bars, 50 µm in A–C.

Click here for additional data file.

Figure S2 Immunostaining of each retinal cell type of AAV-2 intravitreally injected mouse retina

Immunohistochemistry on the transverse retinal cryosection with intravitreal injection of AAV-2-CAGGS-EGFP for (A–C) rod photoreceptors labelled with rhodopsin, (D–F) bipolar cells labelled with PKCα, (G–I) horizontal cells labelled with calbindin, and (J–L) Mueller cells labelled with GS. INL, inner nuclear layer; ONL, outer nuclear layer. Scale bars, 50 µm in A–L.

Click here for additional data file.

Additional Information and Declarations

Competing Interests

Author Contributions

Animal Ethics

Data Availability

The authors declare there are no competing interests.

Yusaku Katada conceived and designed the experiments, performed the experiments, analyzed the data, contributed reagents/materials/analysis tools, prepared figures and/or tables, authored or reviewed drafts of the paper.

Kenta Kobayashi contributed reagents/materials/analysis tools.

Kazuo Tsubota authored or reviewed drafts of the paper.

Toshihide Kurihara conceived and designed the experiments, contributed reagents/materials/analysis tools, authored or reviewed drafts of the paper, approved the final draft.

The following information was supplied relating to ethical approvals (i.e., approving body and any reference numbers):

Institutional Animal Care and Use Committee of Keio University School of Medicine provided full approval for this research (#2808).

The following information was supplied regarding data availability:

Katada, Yusaku (2018): Evaluation of AAV-DJ Vector for retinal gene therapy. figshare. Dataset. https://doi.org/10.6084/m9.figshare.7205057.v1.

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
