# Peer review of "Evaluation of AAV-DJ vector for retinal gene therapy"

_PeerJ, doi:10.7717/peerj.6317_

## Round 0.1 · original submission · Minor Revisions

Several clinical trials of retinal gene therapy are on going. For safe and effective gene transfer to diseased retinas, advancements in vector design, improvements of delivery routes, and selection of optimal timing for intervention will contribute to success of retinal gene therapy. The authors showed efficient gene transfer with AAV-DJ vector containing CAGGS promotor compared with AAV-2 vector. By intravireal injection, AAV-2 vector showed restricted gene transfer to the retinal ganglion cells and the shallow inner nuclear layer, but AAV-DJ vector reached deeper layers. AAV-DJ vector also resulted in gene transfer into the photoreceptor layer and the inner retinal layers by subretinal injection, while gene transfer with AAV-2 vector was limited to the photoreceptor outer segment. These results indicated efficient gene transfer capability of AAV-DJ vector into retina. It is appropriate to accept this manusript with minor revision pointed out by the reviewers.

Reviewer #1 comments
1. Authors should add more explanation in a few sentences that the fluorescence along the blood vessels was due to Mueller glias protrusions in line 203 and 204.
2. AAV-DJ has lower affinity than that of AAV2 in line 218-219, meaning that heparin binding affinity does not explain higher transduction efficiency of AAV-DJ. The authors should discuss other factors possible to explain higher efficiency of AAV-DJ vector in Discussion, if possible.
3. For comparison, please add transduction data with AAV-2 to Figure 3.

Reveiwer #2 comments,
- For the first comment, please add description in “Results” for Figure 2B and 2D.
-For the second comment on Figure 5, the authors should add damaged control if possible.

Reviewer 1 ·

Basic reporting

There are some issues in this article. It is described below.

1. In the INTRODUCTION, observation of gene delivery to retina by AAV in vitro and in vivo should be referred more because it is obscure what AAV can do in retinal gene therapy.
2. Line 93 and 163; Please correct for the words (CsCl2→CsCl2, DJ→AAV-DJ)
3. There are no results in Figure 2 H and I, but they are described in the legend.

Experimental design

Research question is unclear. The issue of retinal gene therapy with AAVs which were already performed need to be described. And, it is also obscure the reason why AAV-DJ was selected from a variety of AAVs as the subject of this study. This should be described, associated with the issue of retinal gene therapy with AAVs.

Validity of the findings

No comment

Additional comments

The article by Katada et al. evaluated the transduction efficiency of AAV-DJ vector for retina in vivo. The authors demonstrated that AAV-DJ had high transduction efficiency with all types of retinal cells and that infectious toxicity was not observed in ERG. The findings of this study are important and are expected to develop gene therapy with AAV-DJ for retinal diseases.

Major
1. The authors mentioned “the expression along the blood vessels seen in flat-mount retinas also was thought to be Mueller glia’s protrusions” in line 203 and 204. But, the evidence was not fully noted.

2. AAV-DJ has a heparin binding domain slimier to AAV-2. The cell tropism of the virus is affected by a binding ability to cell surface. However, figures 2 and 3 showed that GFP expression in Mueller cells, bipolar cells, horizontal cells, and rod photoreceptors was only found with AAV-DJ. The discussions on the function of AAV-DJ and AAV-2 or the property of these cells should be described.

3. Figure 3 only demonstrates the immunostaining imaging of retina in mice which were injected with AAV-DJ. To compare cells transduced by AAV-DJ and AAV-2, the data of AAV-2 introduction is essential (e.g. as supplementary information).

Minor
1. To define the data in the text of RESLUT, it should be represented by the number and the alphabet (e.g. Figure 1A).

Reviewer 2 ·

Basic reporting

AAV-DJ (type2/8/9 chimera) has a relatively high transduction efficiency of interest gene as a powerful tool applied for gene therapy. In this study, the authors made a successful EGFP gene transferring into retina of C57BL/6 mice. As the authors described, the results highlighted the potential usage of AAV-DJ for retinal disorders. Good news is that the neuron signal transferring pathway shown by ERG waveforms was not affected by AAV-DJ expression in retina.
These results give us a supporting data for application of AAV-DJ vector for retina-associated diseases.

Experimental design

To compare transduction efficiency between AAV-DJ and AAV-2, especially for RGC, lower amount of the vector (unsaturated dose) may be desirable in Fig. 1.

Validity of the findings

In this study, reasonable results and conclusions were acquired.

Additional comments

major comments

In Figure 2B (AAV-2), the bottom layer of ONL is intensively stained by DAPI, while there was no strong staining at the bottom of ONL in Figure 2D (AAV-DJ), rather the ONL was stained evenly by DAPI, and the bottom layer had partially fluorescence brought by AAV-DV. The authors should describe these structural difference and partial fluorescence at the bottom of ONL.

In Figure 5, the authors measured ERG waveforms, and showed that EGFP expression did not affect the function. There was no damaged control. The authors should add another control if possible.

minor comments

All figures referred in the text should be specified like Fig. 1A, B not Figure 1.

In Figure 1 legend for D and E, the authors should describe in the legend when these confocal images were taken postinfection.

Correct the reference format in the text in line 90,
(Hitoshi, Ken-ichi & Jun-ichi, 1991) to (Niwa, Yamamura & Miyazaki, 1991). Please check other references as well.

Do not use letter “4oC” instead of “4˚C” (line 126).

In the title of Figure 2 legend, “transfection” should be changed into suitable word.

In Figure 2 legend in line 4, (C-I) should be (C-G), otherwise missing Figure 2H and Figure 2I should be added, and the authors should describe them in the text.

---

## Round 0.2 · accepted · Accept

It should be noted that the legends of Fig. S1 and Fig. S2 are missing in the updated files, although they were found in the rebutal letter (peerj-31959-Rebuttal_Letter_DJ_PEERJ_20181119).

Please confirm to upload the legends on Fig. S1 and S2.

# Reviewer 1 ·

Basic reporting

No comment

Experimental design

No comment

Validity of the findings

No comment

Additional comments

The authors corrected the issues which I pointed out.
I expect that this paper will be accepted.

Reviewer 2 ·

Basic reporting

No comment

Experimental design

No comment

Validity of the findings

No comment

Additional comments

The revised manuscript is clearly enough for understanding smoothly the experimental design and results.
This is a prospective study that rAAV-DJ particles could pass-through photoreceptors with intravitreal injection and apply for retina-associated diseases possibly.